# A Review of the Current State and Future Prospects in Resource Recovery of Chinese Cereal Vinegar Residue

**DOI:** 10.3390/foods11203256

**Published:** 2022-10-18

**Authors:** Ke Wang, Yongjian Yu, Shuangping Liu, Yuanyuan Zhu, Peng Liu, Zhen Yu, Yuqin Wang

**Affiliations:** 1School of Grain Science and Technology, Jiangsu University of Science and Technology, Zhenjiang 212100, China; 2National Engineering Laboratory for Cereal Fermentation Technology, State Key Laboratory of Food Science and Technology, School of Food Science and Technology, Jiangnan University, Wuxi 214122, China

**Keywords:** vinegar residue, resource recovery technologies, anaerobic digestion, feed, fertilizer, high-value products, soil/water remediation

## Abstract

Vinegar residue (VR) is a typical organic solid waste in Chinese cereal vinegar production. It is characterized by high yield, high moisture and low pH and is rich in lignocellulose and other organic matter. To avoid the environmental pollution caused by VR, it should be properly treated. The industry’s existing treatment processes, landfills and incineration, cause secondary pollution and waste of resources. Therefore, there is an urgent demand for environmentally friendly and cost-effective resource recovery technologies for VR. To date, a considerable amount of research has been performed in the area of resource recovery technologies for VR. This review summarizes the reported resource recovery technologies, mainly anaerobic digestion, feed production, fertilizer production, high-value product production and soil/water remediation. The principles, advantages and challenges of these technologies are highlighted. Finally, as a future perspective, a cascade and full utilization model for VR is proposed by considering the inherent drawbacks and economic-environmental feasibility of these technologies.

## 1. Introduction

Vinegar is one of the most popular acidic condiments worldwide. In China, cereals (sorghum, wheat bran, rice, millet) are usually used as raw materials to produce vinegars with characteristic flavors [1]. Traditional cereal vinegars are commonly produced by solid-state fermentation (SSF). The production process is shown in Figure 1 [2]. Cereal starch is converted to fermentable sugars by the action of enzymes contained in the fermentation starter-Qu and is further fermented to alcohol. The alcohol mash is then mixed with wheat bran, rice hull and vinegar starter-Pei to start acetic acid fermentation (AAF). The microorganisms contained in Pei ferment alcohol into acetic acid and produce various flavors during approximately 20 days of AAF. After AAF, vinegar is leached out, and the residual solid component is vinegar residue (VR), which is a main organic solid waste of the vinegar industry. There are many different kinds of vinegars in China, among which Zhenjiang aromatic vinegar, Shanxi aged vinegar, Sichuan bran vinegar and Fujian Monascus vinegar are the most famous [3]. Due to the different raw materials and production processes of these vinegars, their VRs show some degree of variation (Table 1). In China, more than 3 million tons of VR are produced annually. Since VR has high moisture, low pH and is rich in lignocellulose and other organic matter, it could result in serious environmental pollution if not be properly treated before discharge. In addition, these characteristics of VR make it difficult to be treated. At the scale of production, the main disposal methods of VR are landfill and incineration. These methods not only cause secondary pollution to air and soil but also represent a waste of bioresources [4]. Therefore, there is an urgent need to develop alternative disposal methods for VR.

Chinese cereal VR contains various microbial metabolites and unutilized components from raw materials during vinegar fermentation, including starch, protein, fiber, lipids, organic acids and inorganic salts (Table 1) [4,5,6,7,8,9,10,11,12,13,14,15,16,17,18,19,20,21,22,23,24], hence having a high resource recovery value. To maximize the disposal capacity and resource recovery value, it is important to develop and promote the use of environmentally friendly and cost-effective VR resource recovery technologies.

At present, resource recovery technologies for VR can be classified into five main categories: (1) anaerobic digestion, (2) feed production, (3) fertilizer production, (4) high-value product production and (5) soil/water remediation (Figure 2). The purpose of this review is to present a survey of these technologies, analyzing their advantages and challenges. As a prospect for the future, a cascade and full utilization model of VR is proposed to achieve optimal resource recovery.

## 2. Resource Recovery Technologies of VR

### 2.1. Anaerobic Digestion

#### 2.1.1. Advances in Anaerobic Digestion Research

Anaerobic digestion has already been extensively applied for the treatment of organic wastes such as sewage, food waste, energy crops and other biomass to produce biogas or organic acids [40,41,42,43,44,45,46,47,48]. VR is one type of organic waste with a carbon-to-nitrogen ratio (C/N) of 15–30, making it appropriate to be treated by anaerobic digestion. Anaerobic digestion processing can solve the problem of pollution caused by VR, and it can also yield a high energy value, so it is one of the most frequently used resource recovery technologies for VR (Figure 3). As a potential substrate of anaerobic digestion, it is necessary to assess the biodegradability and biogas yield performance of VR. In general, biochemical methane potential (BMP) assays are applied to assess the biodegradability and biogas potential of anaerobic digestion [49,50]. Through the BMP assay, the highest cumulative methane yield from VR was found to be 242.69 mL/g volatile solids (VS) at a feed to inoculum ratio (F/I) of 1 [8], indicating that anaerobic digestion is a promising method for VR treatment. Anaerobic digestion of VR is a complex and multistep process that involves many different microorganisms. First, macromolecules are hydrolyzed and fermented by fermenting microorganisms to produce volatile fatty acids (VFAs) and alcohols (hydrolysis/acidification process). Then, acetic acid-producing bacteria convert VFA and alcohol into acetic acid, carbon dioxide and hydrogen (synthetic acetic acid-producing process). Finally, methanogenic archaea use acid-producing products to produce methane (methanogenic process). Therefore, there are several important factors that affect the performance of anaerobic digestion, such as the organic loading rate (OLR), temperature and pH. These factors are associated with the suppression of VFA accumulation in methanogenic processes [51,52].

OLR is one of the most important parameters in anaerobic digestion, as the balance between hydrolysis/acidogenesis and methanogenesis is significantly influenced by OLR [53]. At a high OLR, the hydrolysis/acidogenesis rate is higher than the methanogenesis rate, leading to a high concentration of VFA accumulated during the hydrolysis/acidogenesis process and eventually irreversible acidification and collapse of the anaerobic digestion system [54]. Li et al. reported that the highest volumetric methane productivity of 581.88 mL/L was achieved at an OLR of 2.5 g/L/d when a continuous stirred tank reactor (CSTR) was used for anaerobic digestion of VR [4]. Nevertheless, with increasing OLR, there was an accumulation of VFA as well as a decrease in pH, resulting in the failure of the anaerobic digestion system. In another study, the microbial community structure of anaerobic digestion with increasing OLR was evaluated. The optimum OLR was 2.0 g VS/L/d with the maximum methane yield. A higher OLR (2.5 g VS/L/d) caused a decreased diversity of the microbial community and irreversible inhibition of the anaerobic digestion process [5].

To achieve a higher methane yield, the operating conditions in the anaerobic digestion of VR were always optimized. Feng et al. applied response surface methodology (RSM) with a central composite design to optimize the anaerobic digestion of VR [6]. A maximum methane yield of 203.91 mL/g vs. and biodegradability of 46.99% were obtained at an F/I of 0.5, organic loading of 31.49 g VS/L and initial pH of 7.29. This maximum methane yield and biodegradability were enhanced by 69.48% and 52.02%, respectively, compared with the conditions before optimization. Conductive materials have been reported to accelerate and stabilize the conversion of organic substrates to methane via direct interspecies electron transfer (DIET) during anaerobic digestion [55,56]. Therefore, two conductive materials, acetylene black (AB) and hydrochar (HC), were tested to promote the anaerobic digestion performance of VR [9]. The addition of 1.0 g/L HC and AB indeed increased methane production by 50% and 232%, respectively. The higher electron conductivity and specific surface area of AB induced more efficient anabolism in anaerobic digestion compared to HC. Several studies have also highlighted the advantages of codigestion in anaerobic digestion, including improved digestibility due to synergistic effects caused by cosubstrates and process stability [57]. Wen et al. developed a new method for the efficient anaerobic codigestion of corn straw (CS) using VR as one substrate and pretreating acidifier [16]. The organic acid in VR was effective in increasing the hydrolysis rate of lignocellulose in CS. The methane yield from the codigestion of VR and pretreated CS reached 140.48 L/kg VS, which was 35.7% higher than that of the substrate mixture without pretreatment.

It should be noted that the rate-limiting step in anaerobic digestion of VR is the hydrolysis of lignocellulose [58,59]. VR consists mainly of bran, rice husk and other filling materials rich in lignocellulose. The structure of lignocellulose contributes to the low biodegradability of VR during anaerobic digestion. To improve biodegradability, pretreatment of VR is required to degrade the lignin fraction into smaller molecules, making cellulose and hemicellulose more degradable for anaerobic microorganisms [60]. Therefore, some studies on anaerobic digestion of VR have focused on pretreatment methods, including steam explosion (SE), alkali, acid, hydrothermal and ultrasonic methods. SE and hydrothermal pretreatment have been widely used in lignocellulosic biomass pretreatment [61,62,63,64]. They are clean methods because no chemicals are used in the process except water [65,66]. SE pretreatment was found by Feng et al. to disrupt the structure of lignocellulose by removing hemicellulose and lignin and effectively increase methane yield [7]. The highest methane yield of 153.58 mL/g vs. was obtained for the SE-treated VR at 0.8 MPa for 5 min, which was 27.65% higher than that of the control. Ran et al. used a hydrothermal method to pretreat washed VR, and the maximum methane yield was obtained when the hydrothermal temperature was 160 °C [11]. Alkali and acid are effective, economical and simple pretreatment methods [64]. Sodium hydroxide (NaOH) pretreatment has been reported to be efficient in removing the majority of lignin and partial hemicellulose from lignocellulose, thus effectively improving the methane yield of VR. The 3% NaOH-treated VR showed the highest cumulative methane yield, 53.99% higher than that of untreated VR [12]. Acid was effective for the release of protein, cellulose and hemicellulose but not for lignin. Hydrochloric acid released more organic substrates for anaerobic digestion than oxalic acid [15]. Ultrasonic pretreatment is a mechanical and environmentally friendly method that does not require additional chemical agents. By ultrasonic pretreatment, the structure of lignocellulose can be destroyed in a very short time by cavitation formation and subsequent collapse while releasing a large amount of energy. For VR pretreated in the condition of 0.5 W/mL ultrasonic power density and 60 min ultrasound, the methane yield could be 68.7% higher than that of the untreated VR [10].

#### 2.1.2. Challenges in Anaerobic Digestion of VR

(1)The biodegradability is low in the absence of pretreatment, which can lead to a gradual accumulation of lignocellulose in the anaerobic digestion reactor. The accumulation of lignocellulose further causes flushing of activated sludge from the reactor and depresses the performance of anaerobic digestion.(2)VR pretreatment methods also have inherent problems [67]. SE, hydrothermal and ultrasonic pretreatments require high energy consumption. Acid and alkali pretreatment can lead to secondary contamination and equipment corrosion. In addition, inhibitors produced during pretreatment, such as furan derivatives, can inhibit microorganisms in anaerobic digestion systems.(3)The residues remaining after VR anaerobic digestion need further treatment. Therefore, the optimized conditions and pre- and posttreatment of VR anaerobic digestion should be further investigated.

### 2.2. Feed Production

#### 2.2.1. Advances in Feed Production

The shortage of conventional feed such as corn, soybean and wheat has limited the development of the livestock industry in many countries. To meet the increasing demand for livestock products and alleviate the conflict between humans and animals for food, it is important to utilize unconventional feed resources [68]. Unconventional feeds made from agricultural and food industry byproducts are now becoming increasingly popular in ruminant feeding systems because of their competitive price compared to conventional feeds [69,70,71,72]. As a byproduct of cereal vinegar production in China, VR can also be used as a potential feed resource considering its nutritional composition (Table 1). In recent years, it has been applied as a nonconventional feed resource directly or in a fermented form for livestock feeding. Song et al. determined the chemical composition and ruminal degradability of VR to assess its feasibility as a feed for ruminants [20]. VR could not be well digested due to the high lignin and ash content. However, ether extract (EE) and protein contents make VR a suitable additive in ruminant diets. VR contains lactic acid, acetic acid, tartaric acid, malic acid and other organic acids. Studies have shown that these fatty acids have certain biological functions in feed. Lactic acid has been successfully used to soak barley grain to adjust the ruminal energetic state, fermentation mode and inherent immunity and to increase the fat content of milk from dairy cows [73,74]. Acetic acid can be easily assimilated by ruminants and used as an energy source. Malic acid is an essential component of energy metabolism associated with the Krebs cycle and contributes a positive role in ruminal pH regulation, microbial fermentation and mean daily weight increase [75,76]. It has also been reported that organic acids entering the diet as feed additives can reduce nitrogen (N) excretion by animals [77]. Song et al. found that the addition of 40 g VR/kg to the diet of laying hens significantly reduced nitrogen excretion and uric acid nitrogen excretion, thereby reducing nitrogen pollution to water and air [21]. VR in laying hen diets could also alter intestinal pH and pepsin activity in the upper gastrointestinal tract but hardly affect the intestinal microbial community of laying hens.

VR can also be fermented by microorganisms to produce microbial fermented feed (MFF). Through fermentation, the content of protein and various nutrients (such as vitamins and amino acids) is increased, and the antinutritional factors in VR are reduced, especially lignocellulose, which can be decomposed by microorganisms. As a result, the nutritional value, nutrient bioavailability and palatability of the feed can be significantly improved. Saccharomyces, Lactobacillus, Bacillus, Aspergillus and Neurospora are commonly used in MFF production [78,79]. In general, these microorganisms have the advantages of efficient enzyme production, nontoxicity, high protein content in the cell, rapid reproduction, and no significant antagonistic relationship with other beneficial microorganisms [79]. *Neurospora sitophila* is a fungal species that has been certified by the U.S. Food and Drug Administration as safe and edible [80]. It is usually used to ferment sugarcane bagasse, corn stover, wood cellulose and other lignocellulosic biomass to make protein- and carotenoid-rich feeds [81,82,83]. When VR was fermented with *N. sitophila*, the nutritional value (higher protein and carotenoid content) and degradability were significantly improved for mutton sheep as roughage feed compared with unfermented VR. In addition, the degradability of protein by ruminal microflora decreased with fermented VR, resulting in more protein being digested and absorbed postrumen [19]. Aspergillus species are commonly used for the production of phytase [84,85,86,87]. It is well known that phytase in feed can reduce the release of phosphorus from livestock and thus avoid phosphorus contamination [88]. As one of the Aspergillus species, *Aspergillus ficcum* has been used to ferment VR for the production of phytase-rich feed additives in SSF [24,25]. Through the Plackett–Burman (PB) design, steepest ascent path design and RSM, the highest phytase activity reached 98.37 ± 0.85 U/g dry mold. Bacillus species, particularly *B. licheniformis* and *B. subtilis*, are important probiotics for humans and animals [89]. They exhibit high stability to gastric conditions and have antimicrobial, anticancer, antioxidant, and vitamin-producing properties [90]. Hydrolyzed VR has been used in the production of feed additive containing *Bacillus licheniformis*. *Bacillus licheniformis* reached a viable count of 8.25 × 109 CFU/mL with a sporulation rate of more than 80% after 18 h of fermentation [26].

#### 2.2.2. Challenges in Feed Production

Despite the mentioned advantages of using VR to produce feed, there are still some risks and problems that must be addressed considering the economic benefit or animal health.

(1)Wet VR is prone to deterioration, which can result in the production of toxic mycotoxins, especially when transported over long distances without proper storage management. Livestock fed with spoiled VR can exhibit respiratory distress, diarrhea, and other toxicities. Therefore, wet VR should be dried before being transported to feed producers, but this process requires significant energy consumption.(2)The low energy concentration and density of VR makes livestock susceptible to satiety and further reduces feed intake. When livestock (particularly monogastric animals) were fed a high percentage of VR, the low digestibility caused by the high fiber content would speed up the passage of chyme through the intestine, thus reducing nutrient absorption [21,91].(3)Higher levels of alcohol left in VR can also pose a risk of poisoning in livestock. Higher levels of alcohol have been reported to cause metabolic disorders, liver disease and brain damage [92].

### 2.3. Fertilizer Production

#### 2.3.1. Advances in Fertilizer Production

Currently, the widespread use of chemical fertilizers has caused significant environmental impacts, such as deterioration of the soil dynamic balance and groundwater contamination. This has promoted the research and development of organic fertilizers with minimal environmental impact [93]. Biomass waste, such as animal manure, sewage sludge waste and food industrial byproducts, are promising renewable raw materials for organic fertilizer production [94,95,96,97,98]. The good physicochemical properties of VR make it a potential organic fertilizer [99]. However, before being used as fertilizer, VR is always pretreated by composting technology to improve fertilizer efficiency. As an effective technology for the utilization of organic solid waste, composting can improve mineralization and humification [100,101]. The composting process typically consists of three stages, including the mesophilic stage, the thermophilic stage and the maturation stage [102]. Soluble and readily decomposable organic compounds are first metabolized by microorganisms. Then, cellulose, proteins and other macromolecules are broken down into humic acids by thermophilic microorganisms [103]. The thermophilic phase usually lasts for 5 days, and the temperature is maintained above 50 °C, which is effective in reducing pathogens [104,105]. When organic matter is reduced and microbial activity is inhibited by high temperatures, the composting process enters the maturation stage in which mesophilic microorganisms dominate and further decompose the remaining organic matter into humic acids [102]. The composting period of VR is commonly long due to the low pH and high lignocellulose content (Table 1). The composting process of VR always needs to be optimized to improve the efficiency and final quality of compost. Zhao used a stable maturation technique and nutrient structure improvement technique to facilitate the composting process of VR. The results showed that the two-stage fermentation technology could achieve full maturation of VR with a germination index (GI) above 100% after fermentation. The addition of plant ashes could significantly improve the composting process by shortening the period and increasing the nutrient contents [27]. Different types and dosages of additives could lead to differences in the chemical composition, nutrient content, quality and toxicity of compost. Calcium carbonate was found to significantly promote VR decay and improve compost quality [28].

VR compost products can promote the growth of crops. Du reported that VR compost-amended media (VR compost mixed with peat and vermiculite in a 6:3:1 (*v*/*v*) ratio) was beneficial for cucumber growth and could be used as a biological control for Fusarium wilt [29]. VR compost-amended media suppressed Fusarium wilt of cucumber by upregulating the activities of defense-related enzymes and pathogenesis-related proteins and by adjusting the expression levels of stress-related genes [30]. VR compost is also a potential organic substrate to control bacterial wilt of tomato seedlings through inhibiting disease and altering the activity of the soil enzymes and microbial community [31].

#### 2.3.2. Challenges in Fertilizer Production

(1)Gases (such as CO_2_, NH_3_ and N_2_O) are emitted during the composting process. These gases not only cause odor problems but also cause greenhouse issues [106]. In addition, NH_3_ emissions result in an excess of 70% of total nitrogen losses [107]. Therefore, process optimization should be performed to control gas emissions.(2)VR consists of bran, rice husk and other filling materials which contain highly crystalline lignocellulose that is recalcitrant to composting. High lignocellulose content in plant wastes has been reported to elongate the composting time in the composting pile. Therefore, to obtain good performance of composting for VR, a long composting period and a large land demand are needed [108].

### 2.4. High-Value Product Production

#### 2.4.1. Advances in High-Value Product Production

VR can also be converted into high-value products through microbial fermentation or physical–chemical processes due to the characteristics of low cost, abundance and easy availability (Figure 4). At present, VR has been a potential substrate for the production of ethanol, butanol and xylose through fermentation processes. However, VR contains a high level of lignocellulose and therefore requires pretreatment and enzymolysis to hydrolyze lignocellulose into fermentable sugars prior to biotransformation. Liu et al. used NaOH-pretreated VR and simultaneous saccharification and fermentation processes to produce ethanol and xylose [22]. The optimal pretreatment conditions were 2.2% NaOH, solid-to-liquid ratio of 1:11 (*w*/*v*), 63 °C, 80 min and 4.9 IU/g xylanase. As a result, the total sugar yield was 66.1% after pretreatment, and the yields of ethanol and xylose were 319 mg/g and 179 mg/g, respectively. It is important to note that the pretreatment process can produce inhibitors to ferment microorganisms, mainly including furfural, 5-hydroxymethyl furfural and other inhibitors. These inhibitors can prevent microbial growth, substrate utilization and product synthesis, thereby greatly reducing productivity [109]. To resolve this problem, microbial strains with high inhibitor tolerance should be bred. A high inhibitor-tolerant strain, *Clostridium acetobutylicum*, has been generated by atmospheric and room-temperature plasma (ARTP) and used to produce biofuel butanol from hydrolyzed VR [32]. After an optimal two-step SE pretreatment and enzymatic hydrolysis, 19.60 g of glucose, 15.21 g of xylose and 5.63 g of arabinose were obtained from 100 g of VR. At the end of the fermentation, 7.98 g/L butanol was achieved.

In addition to microbial fermentation, VR can also be converted into high-value products through physical–chemical processes. Because of the existence of ordered hydrogen bonds, cellulose was reported to naturally form alterable supramolecular structures with hydrophilic, biocompatible and chiral characteristics. Therefore, region-selective functional derivatives can be prepared from cellulose [110,111]. These species of cellulose derivatives can be applied as catalysts for the preparation of pharmaceutical intermediates and fine chemicals. By using the characteristic of high cellulose content in VR, Qiao et al. constructed a novel structural network of VR sulfates with inorganic/organic sulfo- and sulfoalkyl chemicals for catalytic application [33]. The modulated surface realized a good catalytic effect toward the synthesis of imidazolidine-2,4-dione derivatives from sulfates. VR was also used to produce syngas and phenols through pyrolysis. In addition, Liu et, al. reported that anaerobic digestion as a pretreatment method for VR could enhance the pyrolysis effect. The highest gas yield and lignin-derived phenols could reach to 43.14% and 42.16%, respectively [34].

#### 2.4.2. Challenges in High-Value Product Production

The greatest challenge faced by this resource recovery technology is the high cost of bioconversion into high-value products through microbial fermentation compared to traditional raw materials of starch or fermentable sugars. The high cost is commonly caused by the following reasons:(1)VR pretreatment usually consumes large amounts of energy, chemicals and hydrolytic enzymes, thus greatly increasing the processing cost of the raw materials.(2)VR pretreatment can produce inhibitors to ferment microorganisms, so these inhibitors should be removed from the hydrolysate prior to fermentation, or highly inhibitor-tolerant microorganisms should be generated.(3)Acid and alkali pretreatment results in a large amount of wastewater that should be treated before being discharged.(4)The final concentration of the product at the end of fermentation is relatively lower than that of the conventional raw material, so the cost of product purification is high.

### 2.5. Soil/Water Remediation

#### 2.5.1. Advances in Soil/Water Remediation

Heavy metal pollution of soil and water has attracted worldwide attention due to its serious threat to organisms and its accumulation in biota. High levels of heavy metals in soil and water result from improper treatment of waste, agricultural and industrial activities, and other human activities [112]. Heavy metals can easily accumulate in the body along the soil–crop–food chain and are toxic to wildlife and humans [113]. Therefore, there is an urgent need for effective and environmentally friendly technologies for the remediation of soil and water contaminated with heavy metals. Currently, the application of organic solid waste materials for the remediation of heavy metal-contaminated soil and water has received increasing attention. This process has several benefits: (1) economical solution of organic waste disposal; (2) return of components from organic waste to the biogeochemical cycle; and (3) improvement of soil fertility [114]. In recent years, some researchers have started using VR to produce soil and water amendments. Pei et al. applied a soil amendment consisting of VR, stainless steel slag and weathered coal to immobilize lead (Pb) in the soil [35]. The release of Pb from the soil was restrained, and the adsorption of Pb by plants was mitigated accordingly. VR can also be used as a support matrix for nanoscale zero-valent iron (nZVI), which is ideal for the remediation of heavy metal-contaminated soil and water through oxidation, reduction, precipitation and/or adsorption (Figure 5A) [115]. The use of VR as the support matrix can resolve the problems of low transport capacity, passivation and aggregation faced by nZVI, thus improving the immobilization capacity of chromium (Cr) in soil. The composite material derived from nZVI supported on VR achieved immobilization efficiencies of 98.68% Cr (VI) and 92.09% Crtotal. Almost all the exchangeable Cr was shifted to organic matter-bound and Fe-Mn oxide-bound compounds [36]. VR can also be used as a carrier material for *Bacillus subtilis* to enhance the biodegradation of phenanthrene in aqueous solution. Zhang et al. prepared an immobilization carrier for *B. subtilis* ZL09-26 with different temperature-treated VRs and found that VR dried at 50 °C (VR50) maximally promoted the growth and phenanthrene degradation of *B. subtilis* ZL09-26 [37]. Moreover, VR50 started after one week of use.

VR can also be converted to biochar, which can then be used for the remediation of heavy metal-contaminated soil and water. Biochar is a solid, carbonaceous, porous product produced by the pyrolytic conversion of organic biomass in an oxygen-limited atmosphere [116]. It is characterized by high porosity, large surface area and abundant functional groups. Several researchers have reported that biochar can remediate heavy metal-contaminated water and soil through mechanisms of ion exchange, complexation, redox, precipitation and electrostatic interactions (Figure 5B) [116]. Biochar prepared with VR in an oxygen-limited atmosphere at 700 °C has been proven to be a promising amendment for Cd remediation in water and soil [38]. In addition, ZnCl_2_ could modify VR biochar (VRB) by forming a highly porous and aromatic structure. When the mass ratio of ZnCl_2_/VRB was 1, the ZnCl2-modified VRB reached the highest cadmium adsorption capacity (236.81 mg/g), which was significantly higher than that of the control (9.96 mg/g) [39].

#### 2.5.2. Challenges in Soil/Water Amendment Production

(1)Although VR-supported nZVI has been effectively used for soil and water remediation, exposure to nZVI has harmful effects on humans and the environment [115].(2)Biochar made from VR has great advantages in soil remediation, but the long-term effects of biochar on soil remain unclear. Therefore, to reduce the possible risks associated with biochar, more attention should be given to the long-term effects and risk assessment of biochar on soil. For example, heavy metals immobilized on biochar may be rereleased due to chemical, physical and biological degradation caused by weathering aging [116].

## 3. Conclusions and Future Prospective

The disposal of VR is a challenge for vinegar producers. There is growing concern about the environmental and ecological impacts associated with VR treatment, and therefore, more sustainable practices are needed. To date, several VR resource recovery technologies have been proposed and studied. However, by assessing the economic feasibility and potential environmental impacts, some of these technologies should be further investigated before industrial application. For example, the price of feed made from VR is unlikely to be high due to the low nutritional value of VR. In addition, fresh VR needs to be dried and transported to feed producers, which further increases the cost of feed production. Therefore, the technology of producing feed from VR is not economically feasible. For the technology of producing high-value products with VR, the greatest challenge for industrial applications is also the high production cost due to VR pretreatment and low productivity.

Considering the inherent defects of various technologies, a single technology generally cannot realize the full utilization of VR, so some existing technologies should be combined. Based on the characteristics of the above VR resource recovery technologies and the existing technology and facility conditions of vinegar producers, we propose a cascade and full utilization model of VR, as shown in Figure 6. Fresh VR should be pretreated first to increase the availability of anaerobic digestion and thus improve biogas production. However, pretreatment methods should be further investigated to meet the conditions as shown in Figure 6 [117]. Then, the pretreated VR is treated by anaerobic digestion reactors in vinegar plants, where easily degradable organics are digested by microorganisms to produce biogas. To further improve the performance of anaerobic digestion, several strategies as shown in Figure 6 should be adopted. Bioaugmentation can improve microbial diversity and populations favoring biogas production by inoculating certain microorganisms into the anaerobic digestion system [118]. For pretreated VR, bioaugmentation with cellulose-degrading microorganisms should be carried out. Recently, new strategies have been put into practice to improve the efficiency of bioaugmentation by immobilizing microorganisms on nanoparticles [119,120]. To improve the performance of strains during anaerobic digestion, different genetic and metabolic engineering approaches can also be used to modify key microorganisms [121,122]. Finally, residue from the anaerobic digestion reactor, including the organics difficult to degrade (such as lignocellulose) and excess activated sludge, is treated with the composting process. Similarly, further research as shown in Figure 6 should be conducted on composting processes to improve performance and compost quality [123]. It should be noted that machine learning (ML) has been gradually applied to various areas of organic solid waste treatment processes, such as anaerobic digestion, composting, thermal treatment and landfills [124]. The general principle of ML is to generalize the relationships between input and output variables through inductive reasoning and then to make informed decisions in new situations based on the relationships learned from the empirical data. ML has the advantages of having high predictive accuracy when applied to complex nonlinear problems, saving time and greatly reducing the labor and resource consumption of unnecessary repetitive experiments. These advantages allow us to use ML to improve the anaerobic digestion and composting process for VR treatment.

The biogas produced during the anaerobic digestion of VR can be used as an energy source for vinegar production, composting and further treatment of wastewater. The compost can be used as fertilizer. The full utilization model recognizes the efficient VR utilization of the whole industrial chain from the beginning to the end of VR disposal. In addition, there are some residues such as distiller’s grains (DGS) from Chinese baijiu production and brewer’s spent grain (BSG) from beer production that show similar properties to those of VR. Therefore, the proposed full utilization model could provide a reference for resource recovery of DGS, BSG and other similar residues.

## Figures and Tables

**Figure 1 foods-11-03256-f001:**
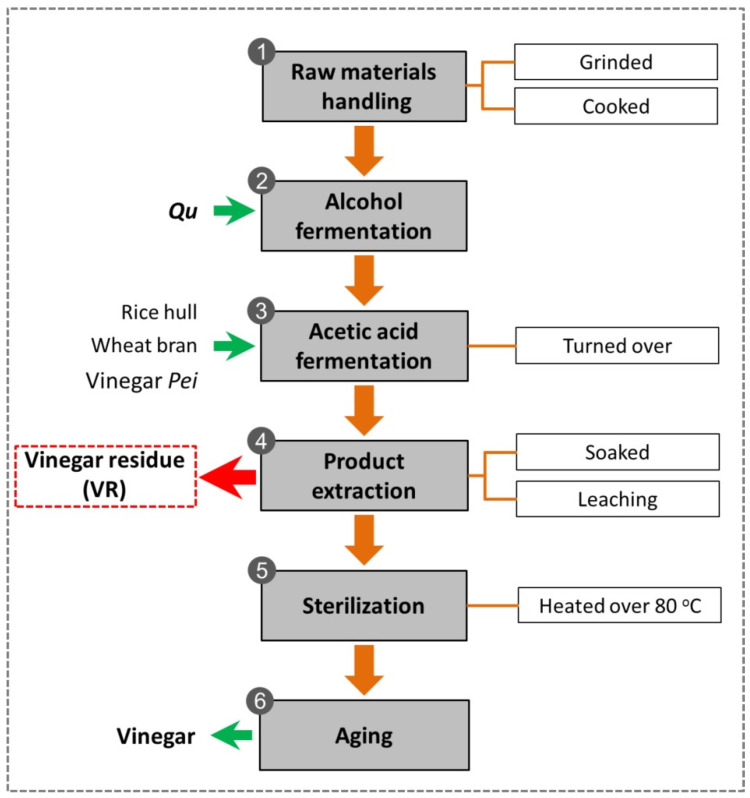
Production process of Chinese cereal vinegar.

**Figure 2 foods-11-03256-f002:**
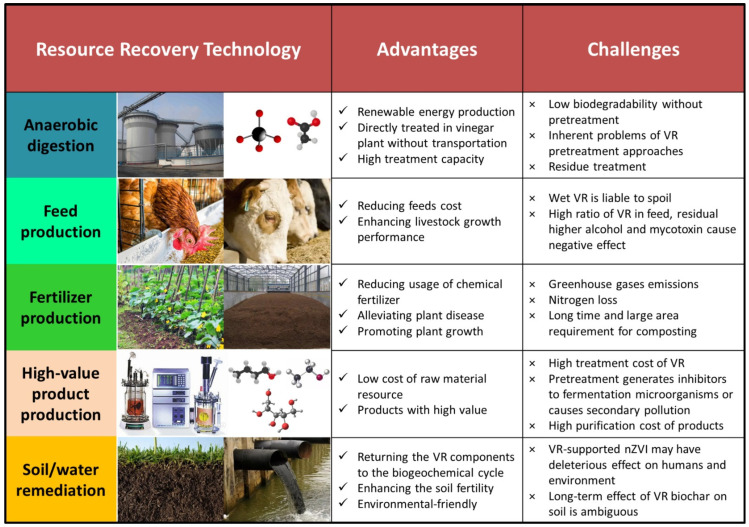
Chinese cereal VR resource recovery technologies [4,5,6,7,8,9,10,11,12,15,16,19,20,21,22,24,25,26,27,28,29,30,31,32,33,34,35,36,37,38,39].

**Figure 3 foods-11-03256-f003:**
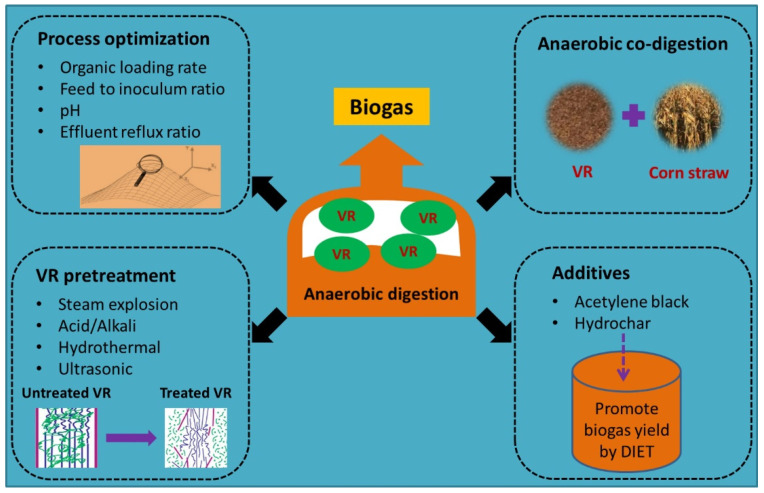
Strategies to improve the performance of the VR anaerobic digestion process.

**Figure 4 foods-11-03256-f004:**
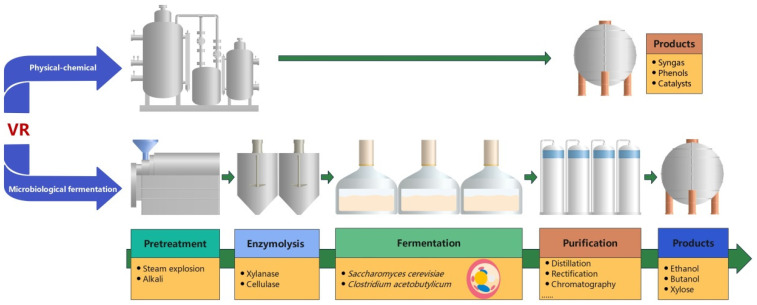
High-value product production from VR through microbiological fermentation or physical–chemical processes.

**Figure 5 foods-11-03256-f005:**
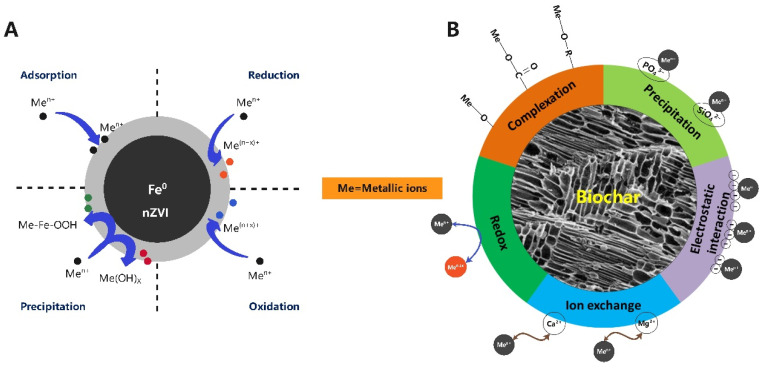
Mechanisms of nZVI (**A**) and biochar (**B**) adsorption and immobilization of heavy metals.

**Figure 6 foods-11-03256-f006:**
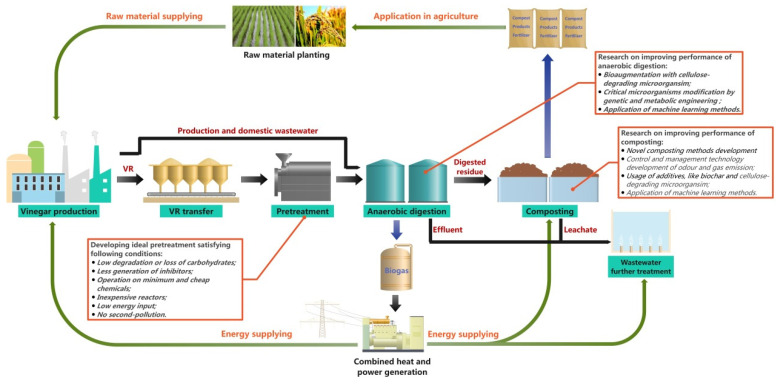
Processes for cascade and full utilization of Chinese cereal VR.

**Table 1 foods-11-03256-t001:** Nutrients and minerals composition of Chinese cereal VR.

Items	Unit	Value
Total solid	%	27.53–35.63
VS/TS	%	88.96–94.73
Crude protein	%TS	9.56–17.60
Crude fiber	%TS	26.70–34.92
Crude fat	%TS	2.70–6.53
Residual starch	k/kg TS	16.12–16.32
Hemicellulose	%TS	16.22–38.90
Cellulose	%TS	22.96–34.91
Lignin	%TS	9.20–24.78
Neutral detergent fiber	%TS	62.40–85.17
Acid detergent fiber	%TS	46.89–55.13
Acid detergent lignin	%TS	22.74–22.74
Ether extract	%TS	5.95–9.98
Acetic acid	k/kg TS	0.15–1.02
Lactic acid	k/kg TS	0.12–1.10
Tartaric acid	k/kg TS	0.16–0.19
Malic acid	k/kg TS	0.04–0.08
Ash	%TS	5.62–13.17
C	%TS	42.14–49.12
N	%TS	1.68–6.61
C/N	NA	15.50–28.68
H	%TS	4.88–6.83
S	%TS	0.08–0.38
O	%TS	35.25–43.44
Calcium	k/kg TS	2.10–2.50
Phosphorus	k/kg TS	0.40–0.63
pH	NA	3.49–4.52

## Data Availability

Not applicable.

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
