# Peer review of "A Review of the Current State and Future Prospects in Resource Recovery of Chinese Cereal Vinegar Residue"

_foods, 2022, doi:10.3390/foods11203256_

Round 1
Reviewer 1 Report
In general, the article has a good structure in the information presented. As general comments, it lacks a justification more focused on the problem or the reason for the possible use of this product. A table could be added, of some already commercial derivative products, if these products are commercialized, what is the potential of their use, in the conclusions, they could be more concrete of the information presented, the Figure supports much of the information.
Author Response
Point 1: As general comments, it lacks a justification more focused on the problem or the reason for the possible use of this product.
Response 1: Justification focused on the problem or the reason for the possible use of this product: Since VR has high moisture, low pH and is rich in lignocellulose and other organic matter, it could result in serious environmental pollution if not be properly treated before discharge. In addition, these characteristics of VR make it difficult to be treated (Lines 79-82, added content). Chinese cereal VR contains various microbial metabolites and unutilized components from raw materials during vinegar fermentation, including starch, protein, fiber, lipids, organic acids and inorganic salts (Table 1), hence having a high resource recovery value (Lines 86-91, original content). The reasons and challenges for each resource recovery technology of VR were shown in section 2.
Point 2: A table could be added, of some already commercial derivative products, if these products are commercialized, what is the potential of their use.
Response 2: Resource recovery technologies for VR summarized in this paper should be further investigated before industrial application, so there are no commercial derivative products produced from VR to date.
Point 3: In the conclusions, they could be more concrete of the information presented, the Figure supports much of the information.
Response 3: By considering the information shown in Figure 6, information presented in the section of conclusions and future prospective has been revised (Lines 462-469, Lines 479-483).

Reviewer 2 Report
A brief summary:
The aim of this review article was to summarize the reported resource recovery technologies of Chinese cereal vinegar residue (VR), especially anaerobic digestion, feed production, fertilizer production, high-value product production, and soil/water remediation. The principles, advantages and challenges of these technologies were emphasized. Furthermore, considering the inherent drawbacks and economic environmental feasibility of the technologies under considerations, as a future issue, a cascade and full utilization model for VR was proposed to achieve optimal resource recovery.
General comments:
In the manuscript, in-depth characterization of the resource recovery technologies of Chinese cereal vinegar residue was presented. Advantages and disadvantages of them were noted. When it comes to challenges of these technologies, which were stated, in some cases there is a need to broaden the description of these subsections. The most crucial is the fact, that in the manuscript the full utilization model, which noted the efficient vinegar residue utilization of the whole industrial chain from the beginning to the end of VR disposal, were presented in the manuscript. It was emphasized that the biogas produced during the anaerobic digestion of VR can be used as an energy source for vinegar production, composting and further treatment of wastewater. Furthermore, it was concluded that the compost can be used as fertilizer. Apart from other significant substantive statements made by the Authors, these are, of course, one of the most important conclusions drawn by them.
Analysing overall merit, the work provides a piece of useful information towards the current knowledge. Hence, this work indeed provides scientific and practical support for the application of the resource recovery technologies of Chinese cereal vinegar residue in the industry. This manuscript is indeed a comprehensive work in this area.
Rating interest to the Readers and taking all the information from this review article into account, in reviewer’s opinion this work for sure can interest the Readers, especially due to their willingness to use up Chinese cereal vinegar residue.
When it comes to the presentation of information, it’s very interesting. What is more, the technologies described in the article are summarized in pictorial figures. The diagrams summarizing the strategies discussed are very purposeful and clearly and concisely present the most important facts. At the same time the whole manuscript contains a lot of useful and interesting information for potential readers. In the reviewer’s opinion the manuscript’s topic is very well presented.
References are well-developed and very exhaustive. Furthermore, the selection of references is also very good.
The specific comments to the manuscript are as follows:
1. In Abbreviations’ section - uneven indentation/ align the indentations before the explanations of abbreviations.
2. I wonder if double interlining is needed in table 1.
3. Lines 222-223 – Authors wrote: ‘ VR contains lactic acid, acetic acid, tartaric acid, malic acid and other short-chain fatty acids’. Why did Authors wrote ‘other short-chain fatty acids’? The listed acids are organic acids and absolutely they are not fatty acids. I wonder if the Authors have put the word ‘other’ here only by mistake, if it were not there, it would not offend the reader. In this situation, it is a major factual error.
4. Subsection 2.3.2. Challenges in fertilizer production (lines 323-327) - the description of these issues should be extended. It’s too briefly mentioned.
5. Line 360 – ‘VR was also used to produce syngas and phenols through pyrolysis’ – this statement should be also expanded to include more details of this experience.
English level:
The language level is generally good.
In my opinion, the significance of the information presented in the manuscript is really high. I have no objection to this article.

Author Response
Point 1: In Abbreviations’ section - uneven indentation/ align the indentations before the explanations of abbreviations.
Response 1: Uneven indentation/ align the indentations before the explanations of abbreviations have been revised (Lines 17-48).
Point 2: I wonder if double interlining is needed in table 1.
Response 2: Table 1 has been revised to single interlining.
Point 3: Lines 222-223 – Authors wrote: ‘VR contains lactic acid, acetic acid, tartaric acid, malic acid and other short-chain fatty acids’. Why did Authors wrote‘ other short-chain fatty acids’? The listed acids are organic acids and absolutely they are not fatty acids. I wonder if the Authors have put the word ‘other’ here only by mistake, if it were not there, it would not offend the reader. In this situation, it is a major factual error.
Response 3: “other short-chain fatty acids” has been revised to “other organic acids” (Line 224).
Point 4: Subsection 2.3.2. Challenges in fertilizer production (lines 323-327) - the description of these issues should be extended. It’s too briefly mentioned.
Response 4: The description of subsection 2.3.2. has been extended (Lines 323-332).
Point 5: Line 360 – ‘VR was also used to produce syngas and phenols through pyrolysis’ – this statement should be also expanded to include more details of this experience.
Response 5: This statement has been extended (Lines 366-369).
